# Real-time multispeckle spectral-temporal measurement unveils the complexity of spatiotemporal solitons

Yuankai Guo[1,2], Xiaoxiao Wen[1,2], Wei Lin[1,2], Wenlong Wang[1], Xiaoming Wei [1✉] & Zhongmin Yang[1✉]

The dynamics of three-dimensional (3D) dissipative solitons originated from spatiotemporal interactions share many common characteristics with other multi-dimensional phenomena. Unveiling the dynamics of 3D solitons thus permits new routes for tackling multidisciplinary nonlinear problems and exploiting their instabilities. However, this remains an open challenge, as they are multi-dimensional, stochastic and non-repeatable. Here, we report the real-time speckle-resolved spectral-temporal dynamics of a 3D soliton laser using a single-shot multispeckle spectral-temporal technology that leverages optical time division multiplexing and photonic time stretch. This technology enables the simultaneous observation on multiple speckle grains to provide long-lasting evolutionary dynamics on the planes of cavity time ($t$) – roundtrip and spectrum ($\lambda$) – roundtrip. Various non-repeatable speckly-diverse spectral-temporal dynamics are discovered in both the early and established stages of the 3D soliton formation.

[1] School of Physics and Optoelectronics; State Key Laboratory of Luminescent Materials and Devices; Guangdong Engineering Technology Research and Development Center of Special Optical Fiber Materials and Devices; Guangdong Provincial Key Laboratory of Fiber Laser Materials and Applied Techniques, South China University of Technology, 381 Wushan Road, Guangzhou 510640, China. [2] These authors contributed equally: Yuankai Guo, Xiaoxiao Wen, Wei Lin. ✉email: xmwei@scut.edu.cn; yangzm@scut.edu.cn

L ightwaves generated and propagated in higher dimensions have recently gained great interest for generating coherent three-dimensional (3D) light fields, delivering high-capacity information and exploring multi-dimensional nonlinear dynamics that widely exist in physics, chemistry, biology, and materials science[1–6]. Lightwave dynamics in multimode waveguides, in particular, have been intensively investigated for their multi-dimensional complexities analogous to other nonlinear systems[2,4–6]. So far, various nonlinear dynamics in multimode fibers have been studied, such as accelerated nonlinear interaction[7], octave supercontinuum generation[8–10], dispersive wave generation[7–9], spatial beam self-cleaning[5], intermodal nonlinear mixing[11], and self-organized instability[12], to name a few. More recently, the 3D soliton—a kind of localized wave with particle-like properties—has also been successively discovered in the multimode fiber[13] and the spatiotemporal mode-locking (STML) laser[14], wherein many transverse and longitudinal modes are simultaneously synchronized to generate 3D femtosecond (fs) solitons. The concept of 3D soliton opens new possibilities for generating extremely energetic fs pulses, manipulating spatio-temporal light fields, and studying higher-dimensional complexities. In spite of these fascinating opportunities, the understanding of the 3D soliton formation is still in its infancy. The presence of multiscale disorders owing to spatiotemporal dispersions and strong nonlinear interactions gives rise to challenges in both theoretical and experimental investigations. Theoretical models have recently been developed for understanding the generation of 3D dissipative solitons[15,16]. An experimental framework for real-time observation, however, is yet to be demonstrated, whereas the time-averaged measurements of 3D solitons have recently been demonstrated by sampling the laser beam profiles[17,18] and manipulating the pump power[19].

Evolving from stochastic seeds in multiple dimensions, 3D soliton dynamics manifest much more complicated behaviors, which are usually unpredictable, time-varying, and non-repeatable. As a result, probing the dynamics in the early and established stages of 3D soliton formation and decomposing the multi-dimensional complexities are the key insight into various higher-dimensional physical and other cross-disciplinary problems, especially those are not experimentally straightforward, e.g., Bose–Einstein condensates, plasmas, polymers, and fluids[20]. Notably, increasing efforts have been made for the real-time characterization of one-dimensional (1D) optical dynamics and interesting transient phenomena have been studied[21–31], e.g., the breathing of dissipative solitons[31], the internal motion of dissipative soliton molecules[27], and the explosion of solitons[22,30]. The real-time observation on 3D soliton dynamics, however, has been largely unexplored, and applying traditional technologies to 3D soliton dynamics is not straightforward[32–36]. Spatiotemporal technologies, such as delay-scanning off-axis digital holography[15], TERMITES[34], SEA TADPOLE[37], and other counterparts[38,39], have recently been demonstrated to study 3D femtosecond pulses with high temporal resolutions—powerful tools for the characterization and optimization of ultrashort pulse lasers. Rather than high repetition rate pulse lasers, they are more suitable for low repetition rate pulse lasers with the identical pulse-to-pulse property. In the meantime, single-shot imaging technologies have also been presented for the real-time observation on 3D lightwave phenomena, e.g., STRIPED FISH[40], STS-CUP[41], CUST[42], to name a few, which, however, have limited numbers of continuous frames and thus prevent the observation of pulse-to-pulse dynamics of 3D solitons that can last for a long period of time. In this work, we present speckle-resolved spectral-temporal dynamics of a 3D soliton laser in real time using a single-shot multispeckle spectral-temporal (MUST) technology, which enables speckle-resolved spectral-temporal observation over a large number of roundtrips (RTs). The speckle-resolved spectral-temporal decomposition of complex multi-soliton dynamics establishes a perfect knowledge of the 3D soliton formation, which sheds new light on understanding the physical nature of 3D dissipative solitons and exploiting their complex instabilities.

## Results

**Principle of MUST.** Figure 1a illustrates the schematic diagrams of the 3D soliton laser and MUST measurement system. The cavity of the 3D soliton laser mainly comprises of 5 m few-mode gain fiber, 2 m few-mode passive fiber, and 2.5 m multimode grade-index (GRIN) fiber. Such a long cavity length (~10 m) can impart a significant nonlinear phase accumulation[43]. To excite higher-order modes, a core offset of ~20 μm is applied to the fusion splicing connection between the few-mode gain fiber and multimode GRIN fiber, where a low-order mode can be converted to higher-order ones (inset of Fig. 1a). The multimode fiber laser is pumped by a multimode laser diode. The mode-locking (ML) operation is accomplished by an intensity-dependent transmission (IDT) mechanism in a nonlinear polarization evolution scheme[44]. Figure 1b, c respectively present the typical pulse train and pulsewidth distribution over speckle grains (SGs) of the 3D soliton laser. A low saturation intensity of the IDT function can allow multipulse STML[18,19], which results in fruitful multipulse dynamics.

The 3D laser beam is extracted by a cubic beam splitter (BS), and subsequently magnified by a ×5 magnification telescope (MT), such that the individual SG can be well resolved. The magnified laser beam is launched to the MUST measurement system, where the spectral-temporal signals of multiple SGs are simultaneously collected by different single-mode probes (SMP$_{1-n}$, here $n = 3$), which can be tapered fibers or fiber collimators (the latter in this study). Please note that the single-mode operation of the probes is crucial for the success of photonic time stretch, also known as dispersive Fourier transform[21]. Adopting the optical time division multiplexing (OTDM) technology widely used in telecommunication, the signals from the SMPs are temporally multiplexed using different lengths of optical delay lines. This step is very important for the robust MUST measurement, otherwise an individual time-stretch channel is required for each SG, leading to a complicated and costly system. The temporally multiplexed signal is then split into two branches by a fiber optical coupler, one of which is directed to a high-speed photodiode (PD) for real-time observation in the temporal domain ($t$). The other branch is passed through a time-stretch unit (~−0.3 ns/nm dispersion) and then detected by another high-speed PD for real-time spectroscopy ($\lambda$). The signals from the PDs are simultaneously recorded by a multi-channel real-time oscilloscope at a sampling rate of 80 GS/s. The recorded data are finally off-line processed and reconstructed to visualize the MUST dynamics of 3D solitons over RTs (top right panel of Fig. 1a). More details about the experimental system are provided in Supplementary Note 1 and Methods.

**Birth dynamics of the multipulse STML.** Very recently, numerical simulations reveal that self-starting dynamics of 3D solitons can be very diverse, as the 3D light field evolved from noise seeds is dominated by different pulse-shaping mechanisms[15]. The MUST observation on the birth dynamics of 3D solitons, on the other hand, holds promise for directly gaining insight into the physical nature of multi-dimensional complexities of 3D solitons. In the experiments, the multimode fiber laser could transit from the continuous-wave (CW) to ML regime when the pump power increases to ~6 W. During the transition from CW to ML regime, the overall spatial mode profile kept

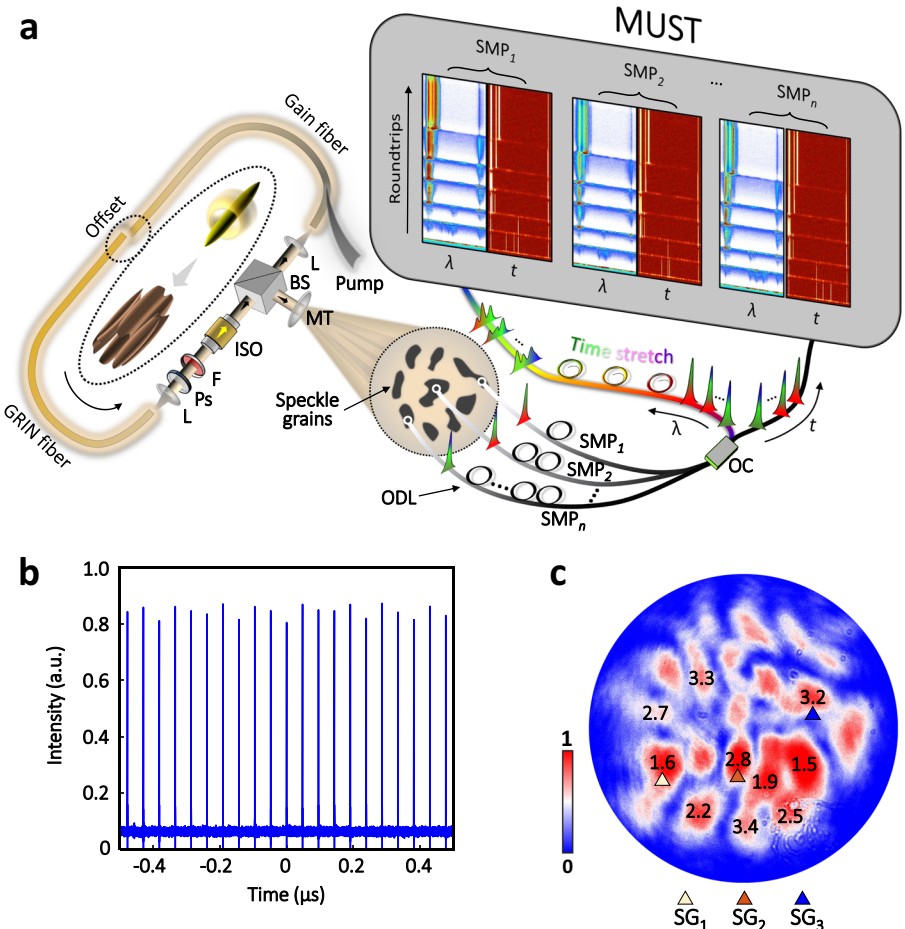

**Fig. 1 Schematic diagram of the real-time MUST measurement system and typical characteristics of the 3D soliton laser. a** Schematic diagram of the MUST measurement system. A spatiotemporal mode-locking (STML) multimode fiber laser is utilized for the study of 3D soliton dynamics. The state of polarization of the laser cavity is adjusted by half-wave and quarter-wave plates (Ps). The STML operation is realized by an intensity-dependent transmission mechanism[14]. The laser beam (i.e., signal) is extracted by a beam splitter (BS), after which it is enlarged by a magnification telescope (MT, 5×) and visualized by the MUST measurement system. In the MUST system, real-time spectral-temporal signals from multiple speckle grains are individually collected by different single-mode probes (SMPs). The collected signals are then temporally multiplexed using optical delay lines (ODLs). The multiplexed signal is split into two branches, one of which is directly detected ($t$). The other branch is launched to a time-stretch unit for real-time spectroscopy ($\lambda$). *F* filter, *ISO* isolator, *L* lens. *OC* optical coupler, *SG* speckle grain. **b** Typical pulse train of the STML laser. **c** Distribution of the pulsewidth (uncompressed) over the speckle grains. Unit: picosecond.

almost consistent except the intensity variation between the bright spots (Supplementary Note 2 and Supplementary Video 1), similar to the results reported in the prior work[14]. A typical MUST landscape of the multipulse STML birth is shown in Fig. 2 (more details are provided in Supplementary Note 3). In this case, the temporal and spectral evolutions over RTs in three different SGs (SG$_{1–3}$, as indicated in Fig. 1c) were simultaneously acquired (Fig. 2a, b). In this shot of multipulse STML birth, different SGs experience a similar landscape that includes relaxation oscillation (RO), Q-switched mode-locking (QSML), and multipulse mode-locking (MPML) states, as indicated in Fig. 2a, and their overall energies follow a similar trajectory (Supplementary Figure 3a).

Despite these common features among different SGs, the spectral-temporal dynamics in each SG are versatile before successful STML. In the RO and QSML states, known as typical lasing behaviors[45], there exist shockwaves in microseconds (μs) with strong intensities (white arrows), i.e., along the RT axis. The front surfaces of the RO shockwaves are calm (Fig. 2b), whereas the rear edge of the first RO shockwave packet carries high-intensity nanosecond (ns) pulses, which subsequently break down to dense pulses in the second RO shockwave (Supplementary

Figure 3a). In the QSML state, in contrast, the calm front surface of the shockwave is gradually disappearing, leaving only these strongly fluctuated dense pulses in the last QSML shockwave. The QSML evolution lasts for ~3700 RTs (i.e., ~180 μs). In each QSML shockwave packet, certain seed pulses (usually the stronger ones) evolve into multipulse trains along the RT axis, and sustain until the next QSML shockwave packet (Fig. 2a). The saturable absorber (SA) effect plays a great role in the QSML dynamics, such that the pulse-shaping mechanism of the SA effect strongly suppresses those weak pulses in the dense pulse cluster[46,47]. After the last QSML shockwave, the oscillator successfully enters the MPML state. Based on the principle of survival of the fittest, certain seeds finally evolve into 3D STML solitons and sustain in the rest of RTs. Interestingly, a new soliton is subsequently born right after STML (yellow arrow in Fig. 2a, and see Supplementary Figure 3e), and coexists with these early generated solitons.

Please note that the time-stretched waveforms of multiple solitons may be temporally overlapped when they are closely located, in which case the main spectral characteristics of the soliton cluster can still be extracted (Supplementary Note 4). In the MPML state, a coherent interference pattern is observed, as

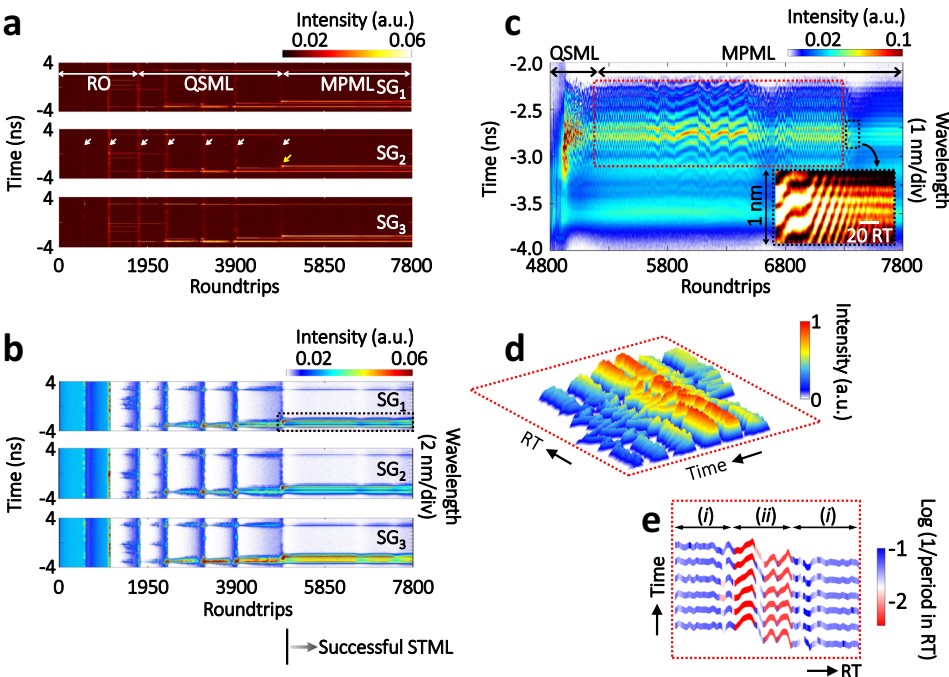

**Fig. 2 MUST birth of the multipulse STML started from successive shockwaves. a** Temporal evolutions of the multipulse STML birth. Here, three speckle grains, i.e., $SG_{1-3}$ as indicated in Fig. 1c, are simultaneously observed. Before successful STML, seven shockwaves have been successively generated (white arrows). After each shockwave, multiple sharp pulses in different patterns are survived or short-lived. **b** Corresponding spectral evolutions. To facilitate the understanding of the wavelength-to-time mapping, the results of the time-stretch measurements in the birth stage are plotted with double Y axis, i.e., time (left) and wavelength (right), which has also been adopted in the prior work[23]. **c** Close-up of the spectral evolution in $SG_1$, indicated by the black dotted rectangle in **b**. Inset exhibits the transient interference fringes resulted from the strong binding effect. The complicated nonlinear interactions between coexisting STML pulses give rise to the complex spectral evolutions. **d**, **e** 3D plot and interference frequency evolution of the area indicated by the red dotted rectangle in **c**. More details have been provided in Supplementary Note 3.

shown in Fig. 2c–e, where the interference fringes are twisted from RT 5100 to 7200 at varying twist frequencies (Fig. 2e). Further study using the field autocorrelation (FAC) technology[24] shows that such a twisted interference pattern can be attributed to the vibration of soliton molecules (Supplementary Figure 3f). Please note that the soliton-molecule in this work is defined as the bound solitons with high coherence, which exhibits high-contrast interference fringes in the spectral domain. The twisted interference pattern experiences two different regimes: (i) RTs of 5100–5800 and 6500–7200, and (ii) RTs of 5800–6500 (Fig. 2e). The regime (ii) is sandwiched by the regime (i), which has a higher twist frequency. The analysis using the FAC technology reveals that these two regimes are also distinguished from each other by their vibration features, wherein the vibration in regime (i) might involve energy exchange between solitons (Supplementary Figure 3f). Such a vibrating state of soliton molecules can be recognized as an excited state of the coherent multi-soliton pattern, which can also decay to the ground state when stronger binding energy is involved[48]. In the final state of the multipulse STML birth, a stable soliton-molecule with temporal separation of ~2.8 ps is generated (Supplementary Figure 3g), evident by the regular interference fringes (inset of Fig. 2c).

As seeded by random multi-dimensional perturbations, the spectral-temporal landscape of the STML birth is largely stochastic from case to case. Figure 3 presents another STML birth with a completely different MUST landscape (more details are provided in Supplementary Note 5). Unlike the former case, here the multiple pulses are simultaneously generated after the strong shockwave, and the spectral-temporal evolution exhibits fruitful multipulse dynamics. Specifically, the multipulse pattern is composed of soliton-molecule with strong binding ($P_1$), soliton

pair without strong binding ($P_2$) and other far-separated ordinary pulses (e.g., $P_3$). As shown in Fig. 3b, c, the stable interference fringes of $P_1$ have a period of ~0.35 THz (angular frequency), corresponding to a temporal separation of ~18 ps. The "likely stable" bound solitons in $P_1$ are further examined by the FAC technology, and the dynamic evolution of the temporal separation between the bound solitons is visualized (Supplementary Figure 7). Moreover, the spectroscopic map of $P_1$ exhibits the energy vibration (Fig. 3d) and dramatic spectral broadening from ~0.5 to ~2.5 nm (Fig. 3e). The solitons in $P_2$ without strong binding, on the other hand, present no obvious interference fringe in the spectral domain (inset of Fig. 3b), which can be attributed to the fact that the density of the interference fringes is beyond the resolving ability of the real-time spectroscopy.

**Soliton-molecule dynamics of the multipulse STML**. We have shown that the spatiotemporal nonlinear interactions between individual solitons can create bound state of 3D solitons—giving rise to 3D soliton molecules. In contrast to the 1D counterpart, e.g., those temporal soliton molecules in single-mode lasers[24], here the 3D soliton molecules with multi-dimensional complexities may share more common features with the original concept of molecules[49] as well as other counterparts existed in biochemistry[50], quantum superfluid and gas[51], etc. As a result, probing the time-varying property of 3D soliton molecules can shed new light on understanding their nature. Figure 4 presents the spectral dynamics of the soliton molecules in $SG_1$ and $SG_2$ that involve time-varying mutual intensity, phases, and separations (also see Supplementary Note 6). In this case, the spectral evolution of the soliton-molecule manifests obvious difference between the SGs. The evolving interference

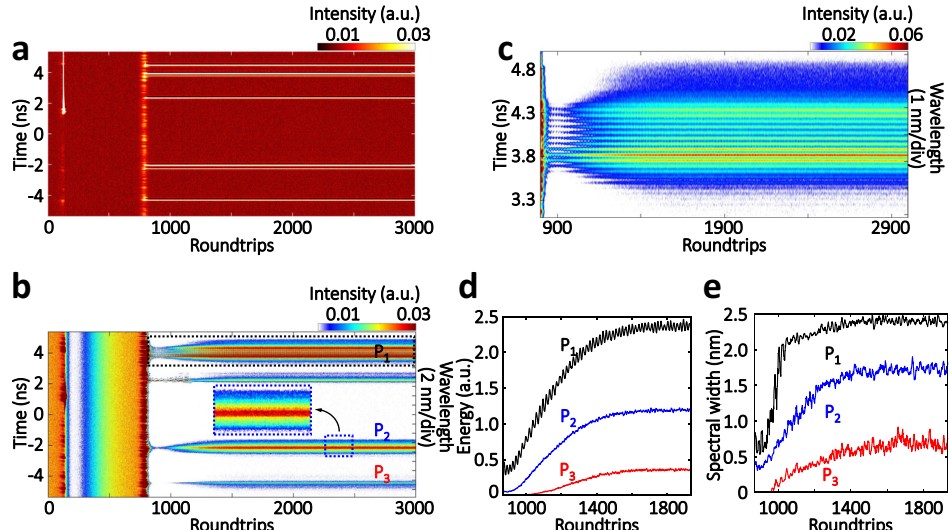

**Fig. 3 MUST birth of the multipulse STML started from a single shockwave. a** Temporal evolution. Here, partial MUST evolution is presented, whereas the full data set has been covered in Supplementary Figure 6. **b** Corresponding spectral evolution. In this case, different pulse clusters coexist in the same roundtrip, including soliton-molecule with strong binding ($P_1$), soliton pair without strong binding ($P_2$) and ordinary soliton ($P_3$). Inset shows the close-up of $P_2$. **c** Close-up of the spectral evolution of the soliton-molecule birth, as indicated in **b**. **d** Energy evolutions of pulses $P_1$–$P_3$. **e** Spectral width evolutions of pulses $P_1$–$P_3$. More details have been provided in Supplementary Note 5.

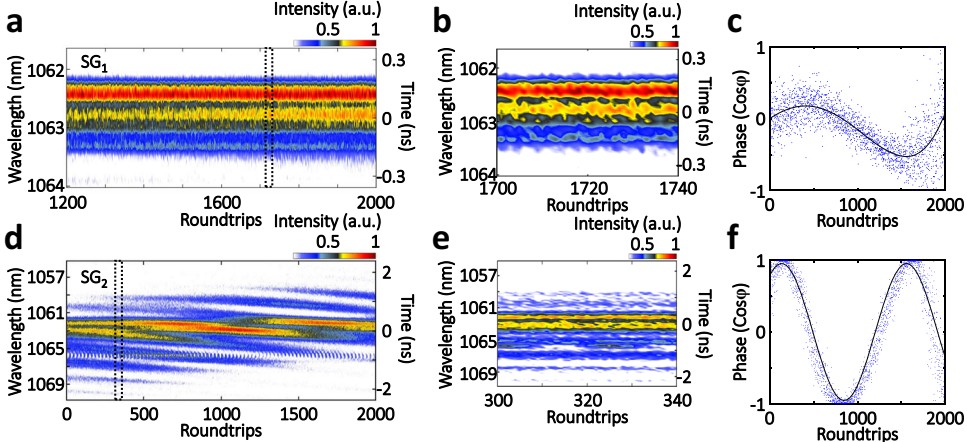

**Fig. 4 Spectral dynamics of 3D soliton molecules. a**, **d** Spectral evolutions in speckle grains $SG_1$ and $SG_2$. **b**, **e** Close-ups as indicated in **a** and **d**, respectively. As can be observed, the soliton molecules in different speckle grains have different spectral widths and temporal separations, which can be identified by the density of the interference fringes. **c**, **f** Phase evolutions of the interference fringes of **a** and **d**, respectively. Please note that, different from the birth stage, here the time-stretch spectroscopy is valid for the whole evolution in the established stage, and thus the "Wavelength" axis is moved to the left for differentiating it from that of the birth stage, the same case for Figs. 5 and 6.

fringes indicate the variations of the temporal separation $\tau$ or relative phase $\Delta\varphi$ of the bound solitons, as shown in Fig. 4c, f (see Supplementary Note 7 for details). In $SG_1$, the relative phase of the bound solitons exhibits a sinusoidal trajectory (Fig. 4c). In contrast, the relative phase in $SG_2$ undergoes a nearly monotonous drift except a weak sinusoidal oscillation (Fig. 4f and Supplementary Figure 9e). These experimental results are complementary to the finding of dissipative bullet molecules, also known as double bullet complex, which has been numerically predicted in the framework of (3+1)-D complex cubic-quintic Ginzburg-Landau equation (CQGLE)[52]. In this regard, the experimental observation on 3D soliton molecules can potentially be a promising platform for unveiling the physics of 3D light bullets dynamics.

**Internal breathing dynamics of 3D dissipative solitons.** So far, the MUST dynamics of 3D solitons have mainly focused on the

interactions between individual solitons, and yet, their internal dynamics of specific solitons—self-evolving over RTs, are also of great interest[15,53,54]. Figure 5a–d illustrate the speckle-resolved spectral-temporal evolutions of such internal pulsating dynamics of 3D solitons in $SG_1$ and $SG_3$. In this case, the mutual interaction between these two far-separated pulses (~3.5 ns apart) can be negligible, and their behaviors are largely independent. Two SGs manifest completely different spectral-temporal landscapes. In the temporal domain, i.e., left panels of Fig. 5e, f, the intensities of the pulses in $SG_1$ oscillate at a high frequency, whereas a low frequency for $SG_3$ (Supplementary Figure 12). Similar properties are also recognized for their intensity integrations (Supplementary Figure 13). In the spectral domain, interestingly, the prominent spectral breathing is visualized in both $SG_1$ and $SG_3$ (right panels of Fig. 5e, f). The evolutions of their spectral energies (Fig. 5g, h), again, oscillate at different frequencies, similar to that of their temporal intensity integrations. The spectral energy oscillations of

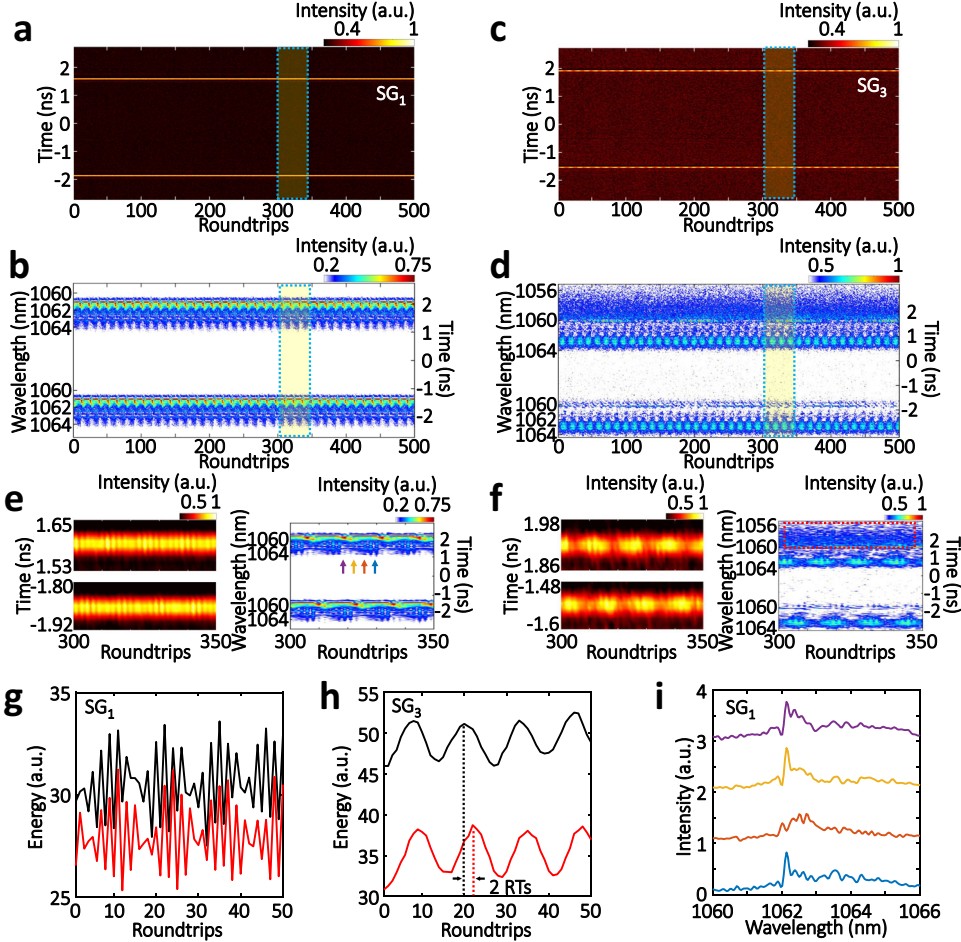

**Fig. 5 Internal breathing dynamics of 3D dissipative solitons. a, c** Temporal evolutions of coexisting two pulses in speckle grains SG₁ and SG₃, respectively. **b, d** Corresponding spectral evolutions. **e, f** Close-ups of the temporal (left) and spectral (right) evolutions of **a, b** and **c, d**, respectively, as indicated in **a**–**d**. **g, h** Energy evolutions in the spectral domain (right panels of **e** and **f**). **i** Snapshot spectra as indicated in the right panel of **e**.

the two solitons in the same SG show a phase delay (~2 RTs, as indicated in Fig. 5h). In addition, the asymmetric spectral dynamics also distinguish SG₁ from SG₃: the upper pulse in SG₃ carries a blue-shifted spectral sideband, as indicated by the red box in Fig. 5f, which does not follow the symmetric spectral breathing like that of the red-shifted sideband. In SG₁, in contrast, both pulses spectrally breathe towards the red side (Fig. 5i). Other spectral landscapes with dual-frequency oscillations are also provided in Supplementary Note 8. To gain a deeper understanding of its fundamental physics, we performed complementary 3D numerical studies (Supplementary Note 9). Two asynchronously pulsating solitons analogous to that of Fig. 5c are obtained in the 3D simulation (Supplementary Figure 18).

The internal soliton dynamics have also been studied for scenarios involving more 3D solitons, as shown in Fig. 6a. In the multipulse cluster, in addition to the ordinary solitons, e.g., P₁ as indicated in Fig. 6b, a spectrally and temporally explosive soliton (i.e., P₂) is unexpectedly observed, as shown in Fig. 6a,c (also their close-ups in Fig. 6b–d). The spectral-temporal explosion of P₂ tends to gradually decay (Fig. 6b). Analogous phenomena of collision-induced explosions with pulsation decay features were obtained in the 3D numerical simulations (Supplementary Figure 22). We also notice in Fig. 6e that P₃ manifests snake-walking spectral evolution with weak spectral width variation (Fig. 6f), while its temporal evolution presents weak intensity modulation (inset of Fig. 6a), which can be attributed to the vibration effect of the soliton-molecule (Supplementary Figure 23).

These results highlight the importance of the real-time MUST measurement for studies of 3D solitons. Specifically, the experimental discovery of 3D soliton pulsation and explosion in the STML platform not only advances the concepts of 1D counterparts, but also offers direct evidence of the pulsating 3D light bullet in the framework of (3+1)-D CQGLE[52,55,56].

## Discussion

To conclude, we have developed a real-time speckle-resolved spectral-temporal observation system to dissect the dynamics of 3D dissipative solitons. As such, we visualized fruitful soliton dynamics in a 3D dissipative soliton laser, where there exist strong spatiotemporal nonlinear interactions among transverse-longitudinal modes—resulting in multiscale disorders. Specifically, various non-repeatable speckly diverse phenomena in both the early and established stages of STML were discovered, including successive shockwaves with decaying front surfaces, evolving soliton molecules with coherent interference pattern, breathing soliton pair with asynchronous internal spectral-temporal dynamics, soliton cluster with spectral-temporal explosion. The macroscopic spectral-temporal properties of 3D solitons in different SGs are mostly consistent in terms of the number of coexisting solitons, the distribution pattern, and the temporal separation. However, because of the complicated spatio-spectral-temporal interactions among the transverse-longitudinal modes, different SGs can exhibit diverse microscopic spectral-

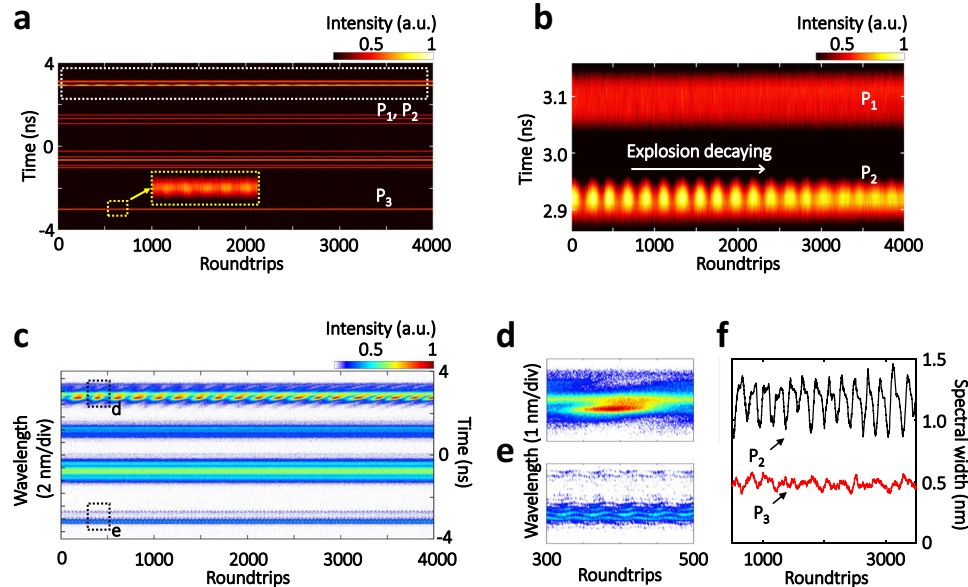

**Fig. 6 Spectral-temporal explosion of 3D dissipative solitons. a** Temporal evolution of the multipulse cluster. Inset shows the close-up of $P_3$. **b** Close-ups of the temporal evolutions of $P_1$ and $P_2$, as indicated by the white dotted rectangle in **a. c** Corresponding spectral evolution. **d**, **e** Close-ups of the spectral evolutions, as predicated in **c. f** Spectral width variations of $P_2$ and $P_3$.

temporal evolutions, which can be determined by the effects of mode coupling and nonlinear interaction, as shown in Figs. 1c, 4, and 5.

The ability of real-time observation on the 3D soliton dynamics is an important step for increased understanding of 3D dissipative solitons, which are also capable of exploring other localized multi-dimensional structures existing in the CQGLE or Gross-Pitaevskii equation[57]. Here, the decomposition of the complex multipulse STML dynamics from their birth to established stages creates a perfect knowledge for understanding the physical nature of 3D dissipative solitons and exploring their complex instabilities. The MUST system can potentially serve as a rich paradigm for studying multidisciplinary problems, e.g., thermodynamics[58], hydrodynamics[59], Bose–Einstein condensates[60,61], etc. It is anticipated that our findings can accelerate the study of these multi-dimensional lightwave dynamics and increase the systematic understanding of their complexities.

Although sampling the mode profile in three SGs can unveil not only the speckly diverse spectral-temporal dynamics but also their pairwise interrelations without loss of generality at a moderate system complexity, simultaneously sampling more SGs might provide a more comprehensive landscape. To illustrate this ability, we have provided an alternative MUST setup for measuring more SGs (Supplementary Note 11), and interesting MUST dynamics have also been observed. In addition, further efforts are yet to be made towards developing single-shot 3D characterization technologies with large numbers of continuous frames for measuring pulse-to-pulse 3D dynamics and fully understanding the nature of their complexities. For example, full-field characterization technologies[25,26] that leverage the time-lens technology[62] can potentially be employed to improve the temporal resolution of the MUST system (Supplementary Table 3).

## Methods

**Experimental setup**. The experimental system consists of a STML multimode fiber laser and MUST measurement system (see Supplementary Figure 1). In brief, the STML multimode fiber laser has a ring cavity, where an Yb-doped fiber (Nufern LMA-YDF-15/130-VIII, 5 m length, 15 μm core size) serves as the gain medium. A multimode grade-index (GRIN) fiber (Thorlabs GIF 625, 2.5 m length, 62.5 μm core size) is fusion-spliced to the gain fiber with a large core offset. A bandpass filter (F) and a polarization-dependent isolator (ISO) are utilized for the STML

operation. The laser signal is extracted by a 50:50 BS. The extracted laser beam is enlarged by a ×5 MT, which is then launched to the MUST measurement system. In the MUST system, the laser signal is split into three branches, and three different SGs of the multimode laser beam are individually received by three fiber collimators. The collected signals are subsequently combined using OTDM. The OTDM signal is split into two parts, one of which is directly detected by a high-speed PD (Newport Model 1544, 12 GHz bandwidth). The other branch is launched to a long single-mode fiber (~−0.3 ns/nm GVD) for real-time spectroscopy. The spectroscopic signal is detected by another PD. The outputs of the PDs are finally recorded by a four-channel real-time oscilloscope at a sampling rate of 80 GS/s.

**Data processing**. First, the OTDM data are segmented according to the RT time of 47.8 ns, resulting in a two-dimensional (2D) matrix $M \times N$, in which each column ($M$) designates temporal (spectral) information, and each row ($N$) represents the RT number. For the time-stretched signal, specifically, a coordinate transform is applied in term of $\lambda = t/D_2$, where $t$ and $\lambda$ represent the retarded time and wavelength, $D_2$ is the amount of dispersion used in the photonic time stretch, i.e., −0.3 ns/nm. Detailed derivation concerning the general case of photonic time stretch is provided in Supplementary Note 4.1. Second, the signals in both temporal and spectral domains are demultiplexed by using the time delays between the SGs, i.e., 15.06 ns and 30.75 ns for $SG_1$-$SG_2$ and $SG_1$-$SG_3$, respectively.

## Data availability
All data used in this study are available from the corresponding authors upon reasonable request.

## Code availability
All custom codes used in this study are available from the corresponding authors upon reasonable request.

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

## Acknowledgements

This work was partially supported by NSFC Development of National Major Scientific Research Instrument (61927816), Guangdong Key Research and Development Program (2018B090904001, 2018B090904003), National Natural Science Foundation of China (NSFC) (U1609219), Local Innovative and Research Teams Project of Guangdong Pearl River Talents Program (2017BT01X137), and Science and Technology Project of Guangdong (2017B030314005).

## Author contributions

X.M.W. conceived the idea. Y.K.G., X.X.W., and W.L. performed the experiments. Y.K.G., X.X.W., and W.L. analyzed the data. Y.K.G. and W.L. conducted numerical simulations. W.L. and X.M.W. wrote the manuscript. All authors commented on the manuscript. X.M.W. and Z.M.Y. supervised the project.

## Competing interests

The authors declare no competing interests.
