## [Peer Review File · Nature Communications]

Reviewers' Comments:

Reviewer #1:

Remarks to the Author:

The manuscript entitled "Real-time multi-dimensional measurement unveils the complexity of spatiotemporal solitons" by Yuankai Guo, Xiaoxiao Wen, Wei Lin, Wenlong Wang, Xiaoming Wei, and Zhongmin Yang addresses the exiting subject of strong current research interests related to the real-time investigation of complex spatiotemporal structures, here the 3D soliton dynamics in, newly emerged, spatiotemporal fiber lasers. Indeed, unveiling and understanding the dynamics of 3D structures is an important question, up to now remaining an open challenge, which calls for development of new advanced multidimensional real-time characterization techniques allowing for measurement of their spatial, spectral and temporal features. In this context, the present manuscript reports the development of a single-shot multispeckle spectral-temporal (MUST) technique, which allowed to measure the real-time non-repeatable speckle-revolved spectral-temporal dynamics of a soliton in both the early and established stages of spatiotemporal fiber laser (STML), including successive shockwaves with decaying front surface, evolving soliton-molecules with coherent interference geometry, breathing soliton pair with asynchronous internal spectral-temporal dynamics, soliton cluster with spectral-temporal explosion. The manuscript is interesting as it provides the first step for understanding the complex 3D dynamics of dissipative solitons, which can provide a further insight in nonlinear multi-dimensional interactions among transverse-longitudinal modes. However, since the measurement technique permits to sample the spatial pattern only in three points I find oversold the main claiming of the manuscript saying that it reports the multi-dimensional measurement of 3D dynamics. From only three points it is difficult to conclude about the dynamics in the whole spatial dimension. Here, it would be very useful to perform also the measurements by using the recently developed, by some of the co-authors, new measurement technique reported in Nat. Comm. 11, 2059, 2020, which would allow to conclude about the real-time spatial dynamics. Moreover, it is important to notice that the dynamics of 3D dissipative solitons has already been measured in few spatial points as reported in the following papers (1) by Ding et al. Optics Express 27, 11435, 2019; (2) by Huanqiang Qin et al. Optics Letters 43, 1982, 2018, although by a standard time-averaged measurement techniques. The authors should refer to those articles in their manuscript. To conclude, I think that the present manuscript is more adapted for other journal than Nature Communication.

Other points that I feel should be addressed are the following:

- 1) Is it possible, basing on performed measurements, to conclude about the relations among the observed dynamics in the three spatial points or those dynamics are independent ?
- 2) Can the authors comment on what was the spatial pattern in the regime of mode-locked? Did it change in comparison to CW or Q-switching regime ? Was it similar to what was reported in Wright et al., Science 358, 94-97, 2017 ?
- 3) I feel that the literature background of the manuscript should be completed by citing also the following two recent papers to allow the reader fully appreciate the results: (1) Peng et al., Sci. Adv. 5, no.11, eaax1110, 2019 which reports the breathing dynamics in mode-locked fiber laser; (2) Krupa at al. PRL 118, 243901, 2017 which demonstrates the first real-time dynamics of internal motion within soliton molecules in fiber laser.
- 4) Can the authors comment on why they reported the DFT measurement as a function in wavelength in the Figs.5 and 6, while as a function of time in Figs.2b, 3b, and 4a,d.

Reviewer #2:

Remarks to the Author:

REVIEW

Title: "Real-time multi-dimensional measurement unveils the complexity of spatiotemporal solitons"

Authors: Y. Guo et al.

In this paper the authors explore real-time speckle and spectral-temporal dynamics in a 3D soliton laser using a single-shot multi-speckle spectral-temporal (MUST) scheme that utilizes optical time division

multiplexing and photonic time stretch methods. In doing so, they investigate successive shockwaves, evolving soliton-molecules, and breathing soliton pairs with asynchronous spectral-temporal dynamics to mention a few. Overall, this work is timely given the rising interest in nonlinear multimode fiber optics - an activity that could lead to not only new phenomena but could also be instrumental in developing new optical high-power sources. The paper is relatively well written and the results are clearly exposed. Having said that however, there still some important technical issues that I would like the authors to seriously address before this paper is considered further.

1. Analyzing an optical wave in space and time, a result that is central in this study, can be performed in a number of ways (like for example interferometrically,) as in the 2016 study of Pariente et al (Ref. 26) where they demonstrated the so-called TERMITES method. Similar methods have also been used by other teams like that of Frank Wise's from Cornell. I think it would be very useful if the authors could elucidate the pros and cons of their scheme compared to other approaches. I honestly found the discussion between lines 57-73 somewhat superficial.
2. It is very hard to tell after reading this paper as to how many modes are indeed involved in this system if one is to declare the solitons observed as "spatio-temporal". The authors should present these results very clearly. From the simulations in the supplementary it will seem that only a handful of modes are actually excited. In that case, what is the power distribution among modes? How does this distribution vary as a function of wavelength and during evolution? These are important aspects that need to be very carefully presented and highlighted.
3. The spatio-temporal structure of these solitons is nowhere to be seen-in way done say in Ref.26. The dynamics are mostly plotted against round-trips. If indeed this scheme is superior to others previously attempted, why is this crucial information missing? I think this crucial information must be provided.
4. Did the authors observe any dispersive wave combs in their experiments? If so can they also present this data?
5. What is the differential group delay for the parabolic fiber used between different propagating modes and is this an issue after thousands of roundtrips?
6. The authors should also mention relevant works in this area like: M. A. Eftekar et al, Nature Communications 10, 1638 (2019), Optics Express 25, 9078-9087 (2017), and Lopez-Galmiche et al, Optics Letters 41, 2553-2556 (2016).

The aforementioned issues must be appropriately addressed before this paper is considered further in Nature Communications.

Reviewer #3:

Remarks to the Author:

In this manuscript, the authors report a new experimental setup to characterize the pulse-to-pulse dynamics of spatiotemporal mode-locked lasers. By combining the dispersive Fourier transform and time division multiplexing, they observe a wide range of ultrafast dynamics. The results are new and interesting. I will support its publication if the authors address the following comments.

- 1) In Fig. 2b and c, I suggest that the authors display the vertical axis with frequency (or wavelength) units if possible. It would make it easier to read the spectral spacing of the interference fringes.
- 2) The inset in Fig. 2c should display units on the axes.
- 3) "In the QSML state [...] the shockwave is gradually disappeared, [...]" I believe that it should read "is gradually disappearing" or "gradually disappeared".
- 4) "After the last QSML [...] enters the continuous MPML state." I think that the term continuous should be removed as it can be mixed up with continuous wave operation.
- 5) I found the sentence "In addition to the ability of real-time [...]" confusing as the experimental setup

records the temporal dynamics without dispersing the signal (Fig. 1a). Moreover, a narrow bandwidth pulse could also have relatively large intensity, especially in Q-switching operating regime.

6) Interference geometry should be replaced by interference pattern.

7) It is not clear to me what the inset in Fig. 3b is showing. I cannot resolve the interference fringes. Is it because the fringe spacing larger than the pulse spectral bandwidth?

8) The authors say that there is only intensity modulation in the time domain on the pulses in SG3, but it looks to me that the intensity of pulses in SG1 are also oscillating at a high frequency which correspond to the oscillations observed in Fig. 5g. Moreover, I would also expect the energy to be the same in the temporal and the spectral domain.

9) In describing Fig.6. the authors say that they observe strong soliton fission. However, I am not really convinced by the experimental results that this is the case. Soliton fission occurs when a fundamental soliton is ejected and temporally separates from a higher-order soliton due to a strong perturbation.

10) Figs. 6c and d are not discussed in the text.

11) While I find the dynamics reported in the manuscript very interesting, I think that this work could benefit from a discussion from the authors on the specificities and limitations of this new MUST setup. For example, could it be improved by adding a time-lens setup?

Manuscript NCOMMS-20-16407-T: “Real-time multi-dimensional measurement unveils the complexity of spatiotemporal solitons”

Authors’ response to reviewers’ comments

We would like to express our gratitude for the reviewers’ valuable comments and suggestions on improving our manuscript. We have taken every comment into consideration and have made appropriate changes to the manuscript. Our point-by-point response appears below, in which we first list the reviewers’ comments in italic letters and then respond to them. The manuscript has also been accordingly revised, and the changes are highlighted in red.

Authors’ point-by-point responses to reviewers’ comments

Reviewer #1: *The manuscript entitled “Real-time multi-dimensional measurement unveils the complexity of spatiotemporal solitons” by Yuankai Guo, Xiaoxiao Wen, Wei Lin, Wenlong Wang, Xiaoming Wei, and Zhongmin Yang addresses the exiting subject of strong current research interests related to the real-time investigation of complex spatiotemporal structures, here the 3D soliton dynamics in, newly emerged, spatiotemporal fiber lasers. Indeed, unveiling and understanding the dynamics of 3D structures is an important question, up to now remaining an open challenge, which calls for development of new advanced multidimensional real-time characterization techniques allowing for measurement of their spatial, spectral and temporal features. In this context, the present manuscript reports the development of a single-shot multispeckle spectral-temporal (MUST) technique, which allowed to measure the real-time non-repeatable speckle-revolved spectral-temporal dynamics of a soliton in both the early and established stages of spatiotemporal fiber laser (STML), including successive shockwaves with decaying front surface, evolving soliton-molecules with coherent*

interference geometry, breathing soliton pair with asynchronous internal spectral-temporal dynamics, soliton cluster with spectral-temporal explosion. The manuscript is interesting as it provides the first step for understanding the complex 3D dynamics of dissipative solitons, which can provide a further insight in nonlinear multi-dimensional interactions among transverse-longitudinal modes. However, since the measurement technique permits to sample the spatial pattern only in three points. I find oversold the main claiming of the manuscript saying that it reports the multi-dimensional measurement of 3D dynamics. From only three points it is difficult to conclude about the dynamics in the whole spatial dimension. Here, it would be very useful to perform also the measurements by using the recently developed, by some of the co-authors, new measurement technique reported in Nat. Comm. 11, 2059, 2020, which would allow to conclude about the real-time spatial dynamics. Moreover, it is important to notice that the dynamics of 3D dissipative solitons has already been measured in few spatial points as reported in the following papers (1) by Ding et al. Optics Express 27, 11435, 2019; (2) by Huanqiang Qin et al. Optics Letters 43, 1982, 2018, although by a standard time-averaged measurement techniques. The authors should refer to those articles in their manuscript. To conclude, I think that the present manuscript is more adapted for other journal than Nature Communication.

Other points that I feel should be addressed are the following:

Authors' response: We are grateful to the reviewer for these valuable comments on improving our manuscript. In the presence of many transverse and longitudinal modes in a spatiotemporal mode-locked multimode fiber laser, the spatiotemporal dispersions and nonlinear interactions can give rise to speckly-diverse dynamics over the multi-speckle mode profile [*Nat. Phys.* 16, 565–570 (2020); *Opt. Lett.* 43, 1982-1985 (2018)], wherein there exist a number of bright spots (i.e., speckle grains), ranging from a few to several tens, determined by the number of coexisting transverse modes of the multimode fiber

laser, e.g., 2–3 bright spots in [*Science* 358, 94–97 (2017)] and ~20 bright spots in the present manuscript. In general, simultaneously probing the spectral-temporal evolutions at more than two bright spots can unveil the speckly-diverse laser dynamics of the multimode fiber laser. From this perspective, sampling the mode profile at three bright spots can unveil not only the speckly-diverse spectral-temporal dynamics but also their pairwise interrelations without loss of generality at a moderate system complexity, e.g., the birth of STML solitons and asynchronous internal spectral-temporal dynamics of breathing soliton pair in **Figs. 2** and **5**, respectively. We also admit that simultaneously sampling more bright spots can provide a more comprehensive landscape, as discussed below.

Scaling of the number of probes: To show the scalability of the number of probes, we have conducted experiments with an alternative configuration to demonstrate the speckly-diverse spectral-temporal dynamics at six spatial spots, as shown in **Fig. R1**. Based on the original design (**Fig. S1**), here another three single-mode probes (SMPs) are added. In brief, the laser signal from the laser cavity is first split into two branches by a 50:50 beam splitter (BS₂). In each branch, the laser signal is further split by two BSs with ratios of 30:70 (BS₃ and BS₅) and 50:50 (BS₄ and BS₆), respectively. The laser signals of three different speckle grains (SGs) in the multimode laser beam are individually received by the SMPs in each branch. Then, the collected laser signals in each branch propagate through different optical delay lines (ODLs), which are subsequently combined by a 3×2 optical coupler (OC) for optical time division multiplexing (OTDM). The OTDM signal is split into two parts in each branch, one of which is directly detected by a high-speed photodiode (PD₁ in the top branch and PD₄ in the bottom branch). The other part is launched to a long single-mode fiber (SMF, Nufern 1060-XP, 8 km length), which provides a large group velocity dispersion (GVD, about -0.3 ns/nm) for real-time spectroscopy through photonic time stretch. Here, the photonic time stretch is performed in a counter-propagation scheme for the two branches. The optical loss is compensated by two Yb-doped fiber amplifiers (YDFAs). The time-

stretched signals of the two branches are individually detected by another two high-speed PDs (PD₃ and PD₂, respectively). The outputs of the PDs are finally recorded by a 4-channel real-time oscilloscope at a sampling rate of 80 GS/s.

Fig. R1. Configuration of the real-time MUST measurement system with six single-mode probes.

BS: beam splitter. C: circulator. Col: collimator. F: filter. ISO: isolator. L: lens. M: mirror. OC: optical

coupler. ODL: optical delay line. PD: photodiode. SG: speckle grain. SMF: single-mode fiber. SPC: signal-pump combiner. YDFA: Yb-doped fiber amplifier. $\lambda/2$: half-wave plate. $\lambda/4$: quarter-wave plate.

Fig. R2. Temporal and spectral evolutions of the multipulse STML at six bright spots. **a.** Temporal (left) and spectral (right) evolutions of the multipulse cluster in speckle grain 1 (SG₁). **b–f.** Temporal and spectral evolutions of the multipulse cluster in SG₂–SG₆, respectively.

As shown in **Fig. R2**, the temporal evolutions of the six bright spots show similar multipulse landscapes in terms of pulse distribution and intensity modulation (mainly the three pulses at the bottom). However, the spectral evolutions of SG_{4–6} exhibit to be different from that of SG_{1–3}, as indicated by the dotted rectangles, wherein prominent spectral breathing is visualized, similar to that of **Fig. 5**. Furthermore, the

spectral evolutions of SG₄₋₆ exhibit different fine structures. In other experiments, the spectral interference with intensity modulation and phase drift using the same measurement system is also observed, as shown in **Figs. R3a** and **b**, respectively.

Fig. R3. Spectral evolution of the multipulse STML. **a.** Spectral evolution of the multipulse STML in SG₃. Inset shows the close-up of the interference fringes. **b.** Single-shot optical spectra at RT 37 and 100, as indicated in **a**.

To address the reviewer’s concern about the scalability of the MUST technology, we have added the scaling of the number of probes to the supplementary information of the revised manuscript, i.e., **Section 11**, and corresponding discussions have also been provided in the **Conclusions and discussion** section:

“Although sampling the mode profile in three speckle grains can unveil not only the speckly-diverse spectral-temporal dynamics but also their pairwise interrelations without loss of generality

at a moderate system complexity, simultaneously sampling more speckle grains might provide a more comprehensive landscape. To illustrate this ability, we have provided an alternative MUST setup for measuring more speckle grains (Supplementary Information 11), and interesting MUST dynamics have also been observed. In addition, further efforts are yet to be made towards developing single-shot 3D characterization technologies with large numbers of continuous frames for measuring pulse-to-pulse 3D dynamics and fully understanding the nature of their complexities. For example, full-field characterization technologies^{25,26} that leverage the time-lens technology⁶² can potentially be employed to improve the temporal resolution of the MUST system (Table S3).” (Conclusions and discussion)

Oversell of the multi-dimensional measurement: In the present manuscript, the multi-dimensional measurement focuses on the simultaneous spectral-temporal observation on multiple speckle grains (bright spots) to unveil the long-lasting evolutionary dynamics on the two-dimensional planes of “cavity time (t) – roundtrip (RT, i.e., related to the discrete distance by $\mathbf{RT} \times l$. Here, l is the cavity length)” and “spectrum (λ) – roundtrip”. From this point of view, it is termed as “**multispeckle spectral-temporal measurement**”. To address the reviewer’s concern, we have carefully revised our manuscript by: 1) changing our title to “*Real-time multispeckle spectral-temporal measurement unveils the complexity of spatiotemporal solitons*”; 2) removing the claim of “*multi-dimensional*” from the manuscript; 3) and defining the multispeckle spectral-temporal measurement:

“Here, we report the real-time speckle-resolved spectral-temporal dynamics of a 3D soliton laser using a single-shot multispeckle spectral-temporal (MUST) technology that leverages optical time division multiplexing and photonic time stretch. MUST enables the simultaneous observation on multiple speckle grains to provide long-lasting evolutionary dynamics on the planes of cavity time

(t) – roundtrip and spectrum (λ) – roundtrip. Various non-repeatable speckly-diverse spectral-temporal dynamics are discovered in both the early and established stages of the 3D soliton formation. These results create intuitive insights into understanding the nature of 3D solitons, and exploring higher degrees of freedom for generating novel 3D lightwaves.” (Abstract).

“In this work, we present speckle-resolved spectral-temporal dynamics of a 3D soliton laser in real time using a single-shot multispeckle spectral-temporal (MUST) technology, which enables speckle-resolved spectral-temporal observation over a large number of roundtrips (RTs).”(Introduction)

Complementary to the spatio-temporal-spectral compressed ultrafast photography (STS-CUP) [Nat. Comm. 11, 2059, 2020]: The STS-CUP technology enables real-time observation on multiple dimensions, i.e., space (xy) + roundtrip (t) or space (xy) + wavelength (λ). Although STS-CUP technology can operate at a speed up to trillions of frames per second, it has a limited total number of frames in each measurement, **i.e., about 60 frames or roundtrips**. This limitation hinders the visualization of dynamic evolutions over long periods of time, e.g., the long-lasting dynamics in both the early and established stages of the spatiotemporal mode-locking, as shown in **Figs. 2–6**, wherein the spectral-temporal dynamics could experience several states (e.g., relaxation oscillation, Q-switched mode-locking and continuous-wave multipulse mode-locking, as shown in **Fig. 2**) and **last for several thousands of roundtrips**. From this point of view, the STS-CUP technology is not suitable for observing long-lasting multi-dimensional optical dynamics — **i.e., the main motivation of the present manuscript to demonstrate a complementary technology to observe the real-time spectral-temporal dynamics of 3D solitons over a large number of roundtrips (or frames) in multiple speckle grains simultaneously**. This limitation of STS-CUP is also applied to another recent work —

the compressed ultrafast spectral-temporal photography (CUST) [*Phys. Rev. Lett.* 122, 193904 (2019)]. In addition, it is technically challenging to combine MUST and STS-CUP to observe the same laser dynamics that evolve over a long period of time, which is because: 1) the STS-CUP technology cannot capture the whole long-lasting evolution as that of the MUST technology; 2) the one-to-one mapping of the results captured by the MUST technology to that of the STS-CUP is difficult, as these two technologies are not temporally synchronized. For these reasons, the STS-CUP technology is not utilized in this work. More discussions have also been provided in the responses to the **Comment 1** of **Reviewer #2** and **Comment 11** of **Reviewer #3**.

To address the reviewer's concern, we have cited these references and appropriately discussed in the revised manuscript:

“Spatiotemporal technologies, such as delay-scanning off-axis digital holography¹⁵, TERMITES³⁴, SEA TADPOLE³⁷ and other counterparts^{38,39}, have recently been demonstrated to study 3D femtosecond pulses with high temporal resolutions — powerful tools for the characterization and optimization of ultrashort pulse lasers. Rather than high repetition rate pulse lasers, they are more suitable for low repetition rate pulse lasers with identical pulse-to-pulse property. In the meantime, single-shot imaging technologies have also been presented for the real-time observation on 3D lightwave phenomena, e.g., STRIPED FISH⁴⁰, STS-CUP⁴¹, CUST⁴², to name a few, which however have limited numbers of continuous frames and thus prevent the observation of pulse-to-pulse dynamics of 3D solitons that can last for a long period of time.” (Introduction)

We have also cited the prior works that demonstrate the time-averaged measurements of 3D solitons by sampling the spatial beam profile in the revised manuscript, i.e., [*Opt. Express* 27, 11435 (2019)] and [*Opt. Lett.* 43, 1982 (2018)], and the corresponding discussion has been provided:

“Theoretical models have recently been developed for understanding the generation of 3D dissipative solitons^{15,16}. An experimental framework for real-time observation, however, is yet to be demonstrated, while the time-averaged measurements of 3D solitons have recently been demonstrated by sampling the laser beam profiles^{17,18} and manipulating the pump power¹⁹.”

(Introduction)

The detailed responses to the other comments are as follows.

Comment 1: *Is it possible, basing on performed measurements, to conclude about the relations among the observed dynamics in the three spatial points or those dynamics are independent?*

Authors’ response: We appreciate the reviewer for the valuable comment. In the real-time MUST measurements, diverse STML soliton dynamics have been observed, e.g., shockwaves formed during the birth of STML solitons, vibrating soliton molecule, breathing soliton pair and spectral-temporal soliton explosion. In the presence of the mode coupling as well as nonlinear modal interaction through Kerr effect and saturable absorption [*IEEE J. Sel. Top. Quantum Electron.* 24, 1-16 (2017); *Science* 358, 94-97 (2017); *Nat. Phys.* 16, 565-570 (2020)], laser dynamics in different speckle grains are considered to be largely correlated in the following ways: 1) STML soliton dynamics in different speckle grains can exhibit similar macroscopic features, particularly in the birth stage of STML solitons. In this case, the mode coupling effect may play a dominating role and take responsibility for their similar macroscopic evolutions; 2) They however show distinctive microscopic spectral-temporal landscapes, for instance, the cases of evolving soliton molecules and breathing soliton pair. In such situation, the nonlinear modal interaction becomes evident, which is associated with transverse mode rotating and temporal drift as demonstrated in the numerical studies (**Supplementary Information 9**). Multispeckle spectral-temporal features of STML solitons can be rather complex, especially in the microscopic scale, which can be

determined by the effects of mode coupling and nonlinear interaction, and we refer this to the principle of minimum loss (maximum gain extraction) [*Nat. Phys.* 16, 565–570 (2020)]. The microscopic speckly-diverse dynamics can also be evident by the pulsewidth variation over the speckle pattern (**Fig. 1c**), particularly when considering the spatiotemporal dispersions. To give more information, the optical spectra of six speckle grains (bright spots) are also measured by using the system shown in **Fig. R1**. As shown in **Fig. R4**, obvious structure variations among the optical spectra of different speckle grains can be observed, in terms of the spectral width and envelope.

Fig. R4. Optical spectra of six speckle grains. The results are measured by using the MUST system with six probes (**Fig. R1**).

To address the reviewer’s concern, we have provided the corresponding discussion in the revised manuscript:

“The macroscopic spectral-temporal properties of 3D solitons in different speckle grains are mostly consistent in terms of the number of coexisting solitons, the distribution pattern, and the temporal separation. However, because of the complicated spatio-spectral-temporal interactions among the transverse-longitudinal modes, different speckle grains can exhibit diverse microscopic spectral-temporal evolutions, which can be determined by the effects of mode coupling and nonlinear interaction, as shown in Figs. 1c, 4 and 5.” (Conclusions and discussion)

Comment 2: *Can the authors comment on what was the spatial pattern in the regime of mode-locked? Did it change in comparison to CW or Q-switching regime? Was it similar to what was reported in Wright et al., Science 358, 94–97, 2017?*

Authors’ response: We thank the reviewer for his/her insightful comments. To address the reviewer’s concern about the evolution of the spatial mode profile when the multimode fiber laser transits from the continuous-wave (CW) regime to the Q-switched (QS) and mode-locking (ML) regimes, we have captured the spatial mode profiles in these three regimes, as shown in **Figs. R5a–c**, respectively. As can be observed, the overall patterns of the spatial mode profiles almost keep consistent except the intensity variation between the bright spots, which is similar to the results reported in [*Science 358, 94–97 (2017)*], i.e., **Figs. 3H, F and J** of the reference. In addition, the spectral and temporal evolutions during the transition from the CW to ML regimes have also been recorded as a function of the pump power, as shown in **Fig. R6**.

To clarify it, we have correspondingly discussed in the revised manuscript:

“In the experiments, the multimode fiber laser could transit from the continuous-wave (CW) to mode-locking (ML) regime when the pump power increases to ~6 W. During the transition from

CW to ML regime, the overall spatial mode profile kept almost consistent except the intensity variation between the bright spots (Supplementary Information 2 and Supplementary Video 1), similar to the results reported in the prior work¹⁴.” (Page 6, paragraph 2)

Fig. R5. Spatial mode profiles of the multimode fiber laser in different regimes. a. Continuous-wave (CW) regime. b. Q-switched (QS) regime. c. Mode-locking (ML) regime. The operating regime of the multimode fiber laser is changed by increasing the pump power, from 5 to 7 W in this case. The dotted circles indicate the intensity variation of the bright spots. Please note that, the setting of the laser cavity in this measurement, mainly the state of polarization, is different from that of the inset of **Fig. 1c**. A video record of the spatial mode profile when increasing pump power is also provided in **Supplementary Video 1** of the revised manuscript.

Fig. R6. Spectral and temporal evolutions during the transition from CW to ML regimes. Please note that, these measurements are time-averaged.

Comment 3: *I feel that the literature background of the manuscript should be completed by citing also the following two recent papers to allow the reader fully appreciate the results: (1) Peng et al., Sci. Adv. 5, no.11, eaax1110, 2019 which reports the breathing dynamics in mode-locked fiber laser; (2) Krupa et al. PRL 118, 243901, 2017 which demonstrates the first real-time dynamics of internal motion within soliton molecules in fiber laser.*

Authors' response: We appreciate the reviewer for this valuable comment on improving our manuscript. Both recent papers by Peng et al. and Krupa et al. have been cited in our original manuscript, i.e., **Ref. 23** and **19**, respectively.

To address the reviewer's concern, the further discussions about these two references (Refs. 31 and 27, respectively) have been provided in the revised manuscript:

“Notably, increasing efforts have been made for the real-time characterization of one-dimensional (1D) optical dynamics and interesting transient phenomena have been studied^{21–31}, e.g., the breathing of dissipative solitons³¹, the internal motion of dissipative soliton molecules²⁷, and the explosion of solitons^{22,30}.” (Page 3, paragraph 1)

Comment 4: *Can the authors comment on why they reported the DFT measurement as a function in wavelength in the Figs.5 and 6, while as a function of time in Figs.2b, 3b, and 4a,d.*

Authors’ response: We appreciate the reviewer for his/her careful studies on our results. In our original manuscript, all the results of the time-stretch (i.e., DFT) measurements are plotted with double Y-axis, i.e., both wavelength and time, which has also been adopted in the prior work [*Nat. Photon. 10, 321–326 (2016)*] (i.e., Ref. 15 of the original manuscript). It can facilitate the understanding of the wavelength-to-time mapping of the photonic time stretch. Please also note that, in **Figs. 2b,c** and **3b,c**, the birth of the STML solitons starts from the quasi-CW regime, in which the photonic time stretch is not valid [*Nat. Photon. 10, 321–326 (2016)*], as discussed in **Supplementary Information 4** of the revised manuscript. For this reason, we place “Time” on the left Y-axis that locates close to the quasi-CW regime of the STML birth, while “Wavelength” on the right Y-axis that locates close to the mode-locking regime. In **Figs. 4–6**, on the other hand, their evolutions are all in the mode-locking regime, and thus we state “Wavelength” on the left Y-axis to differentiate them from that of **Figs. 2b,c** and **3b,c**.

To address the reviewer’s concern, we have elaborated it in the revised manuscript:

“b. Corresponding spectral evolutions. To facilitate the understanding of the wavelength-to-time mapping, the results of the time-stretch measurements in the birth stage are plotted with double

Y-axis, i.e., time (left) and wavelength (right), which has also been adopted in the prior work²³.”

(Caption of Fig. 2)

“Please note that, different from the birth stage, here the time-stretch spectroscopy is valid for the whole evolution in the established stage, and thus the “Wavelength” axis is moved to the left for differentiating it from that of the birth stage, the same case for Figs. 5 and 6.” (Caption of Fig. 4)

We hope that these revisions could satisfy the reviewer’s concerns and that they meet the publication requirements. Thank you very much for your attention and consideration to our paper.

Reviewer #2: *In this paper the authors explore real-time speckle and spectral-temporal dynamics in a 3D soliton laser using a single-shot multi-speckle spectral-temporal (MUST) scheme that utilizes optical time division multiplexing and photonic time stretch methods. In doing so, they investigate successive shockwaves, evolving soliton-molecules, and breathing soliton pairs with asynchronous spectral-temporal dynamics to mention a few. Overall, this work is timely given the rising interest in nonlinear multimode fiber optics-an activity that could lead to not only new phenomena but could also be instrumental in developing new optical high-power sources. The paper is relatively well written and the results are clearly exposed. Having said that however, there still some important technical issues that I would like the authors to seriously address before this paper is considered further.*

Authors' response: We thank the reviewer for the valuable comments on improving our manuscript. The responses to all the comments are presented as follows.

Comment 1: *Analyzing an optical wave in space and time, a result that is central in this study, can be performed in a number of ways (like for example interferometrically,) as in the 2016 study of Pariente et al (Ref. 26) where they demonstrated the so-called TERMITES method. Similar methods have also been used by other teams like that of Frank Wise's from Cornell. I think it would be very useful if the authors could elucidate the pros and cons of their scheme compared to other approaches. I honestly found the discussion between lines 57-73 somewhat superficial.*

Authors' response: We appreciate the reviewer for his/her insightful comments. In STML multimode lasers, coexisting transverse and longitudinal modes can be strongly coupled and nonlinearly interact, leading to complex spatio-spectral-temporal dynamics, particularly when they operate in the partial-STML regime that detunes from the optimal condition, i.e., the cases demonstrated in the present work.

Here, the underlying dynamics at the round-trip level can last for thousands of roundtrips, e.g.,

the spectral-temporal evolutions in both the early and established stages of STML, as shown in **Figs. 2–6** of the present manuscript, wherein the non-repeatable dynamics experience several states (e.g., relaxation oscillation, Q-switched mode-locking and continuous-wave multipulse mode-locking, as shown in **Fig. 2**). From this point of view, spatiotemporal characterization technologies operating at a time-averaged mode or with limited numbers of frames are not suitable for the visualizations of long-lasting dynamics, such as these demonstrated in the present manuscript — i.e., the main motivation of the present manuscript that demonstrates a complementary technology to observe the real-time spectral-temporal dynamics of 3D solitons evolving over a large number of roundtrips (RTs, or frames) in multiple speckle grains. The detailed discussions about different technologies are as follows:

Total E-field reconstruction using a Michelson interferometer temporal scan (TERMITES):

TERMITES is a powerful tool for spatiotemporal characterizations of ultra-intense femtosecond lasers. In TERMITES, a portion of the laser beam under test is used as the reference, and the properties of the laser pulses at all other points of the laser beam are compared with this reference. To reconstruct the spatiotemporal profile of an ultra-intense femtosecond pulse laser, TERMITES must precisely scan the time delay between the two interferometer's arms and record the image of the spatial interference pattern for each delay. **The beauty of TERMITES is that it can provide 3D structures of ultra-intense femtosecond laser pulses, which is significant for characterizing and optimizing ultra-intense ultrashort pulse lasers.** On the other hand, it takes a long time to record the interferograms for reconstructing the 3D profile of the pulse laser, in which period multiple pulses are involved. Consequently, TERMITES might be more suitable for low repetition rate lasers with identical property from pulse to pulse. In this aspect, it is similar to SEA TADPOLE — spatial encoded arrangement for temporal analysis by dispersing a pair of light E-fields [*Opt. Express* 15, 10219–10230 (2007)], as well

as the technology used by Frank Wise's team — delay-scanning off-axis digital holography [*Nat. Phys.* 16, 565–570 (2020)].

Other single-shot technologies: To overcome the time-averaged issue, the STRIPED FISH technology has been proposed for measuring full spatiotemporal fields of ultrashort pulses [*Opt. Express* 14, 11460–11467 (2006); *J. Opt. Soc. Am. B* 25, A25–A33 (2008)]. It employs wavelength-multiplexed digital holography to measure the full spatiotemporal electric field of a single ultrashort laser pulse. To achieve single-shot measurement, STRIPED FISH captures a large digital hologram containing multiple smaller holograms, each of which characterizes the spatial intensity and phase distributions of an individual wavelength component of the laser pulse. **The beauty of the STRIPED FISH technology is that it can measure the full-field information of a single pulse, typically from pulse lasers working at low repetition rates (e.g., sub-kHz).** It is invalid for analyzing multiple sequential high repetition rate pulses, otherwise the dynamics over many pulses are time-averaged in each measurement. For the same reason, the STRIPED FISH technology is not suitable for measuring the dynamics between the round-trip pulses at tens of MHz repetition rates, e.g., the STML multimode fiber laser in our work.

In addition to the STRIPED FISH technology, some other single-shot technologies working at burst mode have also been recently demonstrated, including **STS-CUP** [*Nat. Commun.* 11, 1–9 (2020)], **CUST** [*Phys. Rev. Lett.* 122, 193904 (2019)], **FRAME** [*Light Sci. Appl.* 6, e17045–e17045 (2017)], **SS-FTOP** [*Appl. Opt.* 53, 8395–8399 (2014)], **LIF-DH** [*IEEE J. Sel. Top. Quantum Electron.* 18, 479–485 (2011)], just to name a few. However, they have limited numbers of continuous frames (**Table R1**), preventing the real-time observation of the long-lasting laser dynamics, e.g., these cases demonstrated in the present manuscript.

To clarify it, we have cited all these references and correspondingly discussed them in the revised manuscript:

“Evolving from stochastic seeds in multiple dimensions, 3D soliton dynamics manifest much more complicated behaviors, which are usually unpredictable, time-varying and non-repeatable. As a result, probing the dynamics in the early and established stages of 3D soliton formation and decomposing the multi-dimensional complexities are the key insight into various higher-dimensional physical and other cross-disciplinary problems, especially those are not experimentally straightforward, e.g., Bose-Einstein condensates, plasmas, polymers and fluids²⁰. Notably, increasing efforts have been made for the real-time characterization of one-dimensional (1D) optical dynamics and interesting transient phenomena have been studied^{21–31}, e.g., the breathing of dissipative solitons³¹, internal motion of dissipative soliton molecules²⁷, and the explosion of solitons^{22,30}. The real-time observation on 3D soliton dynamics, however, has been largely unexplored, and applying traditional technologies to 3D soliton dynamics is not straightforward^{32–36}. Spatiotemporal technologies, such as delay-scanning off-axis digital holography¹⁵, TERMITES³⁴, SEA TADPOLE³⁷ and other counterparts^{38,39}, have recently been demonstrated to study 3D femtosecond pulses with high temporal resolutions — powerful tools for the characterization and optimization of ultrashort pulse lasers. Rather than high repetition rate pulse lasers, they are more suitable for low repetition rate pulse lasers with identical pulse-to-pulse property. In the meantime, single-shot imaging technologies have also been presented for the real-time observation on 3D lightwave phenomena, e.g., STRIPED FISH⁴⁰, STS-CUP⁴¹, CUST⁴², to name a few, which however have limited numbers of continuous frames and thus prevent the observation of pulse-to-pulse dynamics of 3D solitons that can last for a long period of time. In this work, we present speckle-resolved spectral-temporal dynamics of a 3D soliton laser in real time

using a single-shot multispeckle spectral-temporal (MUST) technology, which enables speckle-resolved spectral-temporal observation over a large number of roundtrips (RTs). The speckle-resolved spectral-temporal decomposition of complex multi-soliton dynamics establishes a perfect knowledge of the 3D soliton formation, which sheds new light on understanding the physical nature of 3D dissipative solitons and exploiting their complex instabilities.” (Introduction)

In addition, we have summarized their key parameters in **Table R1** (i.e., **Table S3** in the revised manuscript).

Table R1 Typical technologies for 3D lightwave characterization

Technology	If single shot	If burst mode	Temporal resolution	No. of continuous frames
Delay-scanning off-axis digital holography	No	No	NA [*]	1 ^{**}
TERMITES	No	No	NA [*]	1 ^{**}
SEA TADPOLE	No	No	NA [*]	1 ^{**}
STRIPED FISH	Yes ^{***}	Yes	NA [*]	1
STS-CUP	Yes	Yes	500 fs	60
CUST	Yes	Yes	0.1–5 ps	60
T-CUP	Yes	Yes	0.58 ps	350
ISIS CCD	Yes	Yes	10 ns	16
FRAME	Yes	Yes	200 fs	4
LIF-DH	Yes	Yes	88 fs	7
SS-FTOP	Yes	Yes	276 fs	4
MUST	Yes	No	12.5 ps and 47.8 ns ^{****}	>260,000 ^{*****}

^{*}Not provided in the paper.

^{**}Multiple measurements are required for reconstructing the 3D structure.

^{***}Single-shot measurement only for low repetition rate lasers.

^{****}Here, the temporal resolution in the temporal domain is defined by the sampling rate of the real-time oscilloscope (80 GS/s), while the temporal resolution in the spectral domain is defined by the repetition rate of the laser (20.9 MHz), i.e., 12.5 ps and 47.8 ns, respectively.

^{*****}Here, the number of continuous frames is defined by the repetition rate of the laser (20.9 MHz) and memory depth of the real-time oscilloscope (1 Gpts/channel)

Comment 2: *It is very hard to tell after reading this paper as to how many modes are indeed involved in this system if one is to declare the solitons observed as “spatio-temporal”. The authors should present these results very clearly. From the simulations in the supplementary it will seem that only a handful of modes are actually excited. Is that case, what is the power distribution among modes? How does this distribution vary as a function of wavelength and during evolution? These are important aspects that need to be very carefully presented and highlighted.*

Authors’ response: We thank the reviewer for his/her careful studies on our results. Based on the key parameters of the optical fibers provided by the manufacturers, i.e., the numerical aperture (NA) and core radius, the numbers of modes supported by the optical fibers are estimated and summarized in **Table R2**, which has been added to the supplementary information of the revised manuscript (**Table S1**).

Table R2 Estimated numbers of modes of the fibers used in the multimode fiber laser

Fiber	Model number	Core NA	Core radius/ μm	V number	Mode no.*
Lead fiber 1	Nufern LMA-GDF-15/130 0.08/0.46NA	0.08	7.5	3.5	6 [†]
Lead fiber 2	Nufern LMA-GDF-15/130 0.08/0.46NA	0.08	7.5	3.5	6 [†]
Gain fiber	Nufern LMA-YDF-15/130-VIII	0.08	7.5	3.5	6 [†]
GRIN fiber	Thorlabs GIF625	0.275	31.25	51.0	650 [‡]

*Counting the polarization; [†]Mode no. $\approx V^2/2$; [‡]Mode no. $\approx V^2/4$

As discussed in **Supplementary Information 9**, for a moderate calculation time, only 10 transverse modes coexisted in the laser cavity are considered in our numerical studies without loss of generality. A typical intensity distribution among these modes is shown in **Fig. R7**, while the evolutions of their intensity distributions at different wavelengths are provided in **Fig. R8**.

Fig. R7. Intensity distribution among modes at RT 300. Here, the wavelength is 1063 nm.

Fig. R8. Evolutions of the intensity distribution. **a.** 1062 nm. **b.** 1063 nm. **c.** 1064 nm.

To address the reviewer's concerns, we have added these results to the revised manuscript, and correspondingly discussed them:

“An Yb-doped gain fiber (Nufern LMA-YDF-15/130-VIII, 5 m length, 15 μm core size) is pumped by a multimode laser diode (~ 30 W maximum power, 976 nm wavelength) through a signal-pump combiner (SPC) with a multimode pigtail (Nufern LMA-GDF-15/130 0.08/0.46NA, ~ 2 m length, 15 μm core size). A multimode grade-index (GRIN) fiber (Thorlabs GIF625, 2.5 m length, 62.5 μm core size) is fusion-spliced to the gain fiber, where a large core offset is applied to excite the higher-order modes. The numbers of modes supported by the optical fibers used in the cavity are estimated and summarized in Table S1.” (Supplementary Information 1)

“In the simulation, for a moderate calculation time, only 10 transverse modes coexisted in the laser cavity are considered without loss of generality. The second-order dispersions of these modes used in the numerical studies are shown in Fig. S15. After successfully mode-locking, a typical intensity distribution among these modes is shown in Fig. S16, while the evolutions of their intensity distributions at different wavelengths are provided in Fig. S17.” (Supplementary Information 9)

Comment 3: *The spatio-temporal structure of these solitons is nowhere to be seen-in way done say in Ref. 26. The dynamics are mostly plotted against round-trips. If indeed this scheme is superior to others previously attempted, why is this crucial information missing? I think this crucial information must be provided.*

Authors' response: We thank the reviewer for the insightful comment. We agree with the reviewer that the visualization of spatiotemporal structures of STML solitons is especially important for understanding the physical origin of the STML complexity. The TERMITES technology reported by Pariente et al.

(Ref. 26 of the original manuscript) indeed is a powerful tool for the spatiotemporal characterization of femtosecond pulses. To reconstruct the spatiotemporal structure of a femtosecond pulse laser, TERMITES needs to acquire multiple interferograms by precisely scanning the time delay between the two interferometer's arms, leading to the time-averaged measurement. From this point of view, TERMITES is more suitable for low repetition rate lasers with identical pulses. In a partial-STML multimode fiber laser working away from its optimal condition, on the other hand, there exist complex spatio-spectral-temporal dynamics at a high repetition rate (typically tens of MHz) that can last for thousands of frames/roundtrips. To illustrate this, we have used the numerical model described on **Supplementary Information 9** to investigate the pulse-to-pulse spatiotemporal dynamics of a partial-STML multimode fiber laser (**Figs. R9,10**). As can be observed, the spatiotemporal structures of the partial-STML pulses can largely change from RT 500 to 2500 (**Fig. R9**), corresponding to a time interval of $95.6 \mu\text{s}$. In some other cases, they can even change from one roundtrip to another (**Fig. R10**), corresponding to a time interval of only 47.8 ns (i.e., the roundtrip time of the laser cavity).

Fig. R9. Spatiotemporal dynamics in the numerical simulation. a. RT 500. b. RT 1500. c. RT 2500.

Here, the 3D isosurface plots are set to 10% of the peak field intensity.

Fig. R10. Pulse-to-pulse spatiotemporal dynamics in the numerical simulation. a. RT 3150. **b.** RT 3151. **c.** RT 3152. The arrows indicate the spatiotemporal variations. Here, the 3D isosurface plots are set to 10% of the peak field intensity.

As discussed in **Comment 1**, spatiotemporal characterization technologies operating at time-averaged modes or with limited numbers of frames are not suitable for observing the long-lasting pulse-to-pulse spatiotemporal dynamics — i.e., the main motivation of the present manuscript is to demonstrate a complementary technology to study the long-lasting pulse-to-pulse spectral-temporal dynamics of the partial-STML multimode fiber laser. We also admit that, although the MUST technology can enable the observation of the pulse-to-pulse spectral-temporal dynamics for multiple speckle grains simultaneously, further efforts are yet to be made towards developing single-shot 3D spatiotemporal technologies with large numbers of continuous frames for measuring the non-repeatable 3D laser dynamics and understanding their nature.

To address the reviewer’s concern, we have added the numerical results to the revised manuscript (**Supplementary Information 9**), and corresponding discussions have also been provided:

“In addition to the internal spatiotemporal dynamics within 3D solitons, pulse-to-pulse dynamics have also been observed when the multimode fiber laser operates in the partial-STML regime, as shown in Figs. S20,21. These numerical results suggest that single-shot characterization technologies are especially important for unveiling the dynamics of STML multimode fiber lasers and understanding their physical origins.” (Supplementary Information 9)

“Although sampling the mode profile in three speckle grains can unveil not only the speckly-diverse spectral-temporal dynamics but also their pairwise interrelations without loss of generality at a moderate system complexity, simultaneously sampling more speckle grains might provide a more comprehensive landscape. To illustrate this ability, we have provided an alternative MUST setup for measuring more speckle grains (Supplementary Information 11), and interesting MUST dynamics have also been observed. In addition, further efforts are yet to be made towards developing single-shot 3D characterization technologies with large numbers of continuous frames for measuring pulse-to-pulse 3D dynamics and fully understanding the nature of their complexities. For example, full-field characterization technologies^{25,26} that leverage the time-lens technology⁶² can potentially be employed to improve the temporal resolution of the MUST system (Table S3).” (Conclusions and discussion)

Comment 4: *Did the authors observe any dispersive wave combs in their experiments? If so can they also present this data?*

Authors' response: We appreciate the reviewer for this valuable comment. In our experiments, both the original submission and the revision, we did not observe dispersive waves that have been studied in both single-mode fiber systems [*IEEE J. Sel. Top. Quantum Electron.* 24, 1101408 (2018)] and multimode fiber systems [*Nat. Commun.* 10, 1638 (2019)]. This can be mainly due to the fact that a **narrow bandpass filter** (about 4 nm bandwidth, see **Fig. S1**) is placed inside the multimode fiber laser cavity to facilitate the STML operation [*Science* 358, 94–97 (2017)], preventing the generation of the dispersive wave components that usually experience large wavelength shift.

To address the reviewer's concern, we have provided the bandwidth of the bandpass filter in the revised manuscript:

“Two half-wave plates ($\lambda/2$) and two quarter-wave plates ($\lambda/4$) are used for polarization control, which is in conjunction with a narrow bandpass filter (F, Semrock LL01-1064-12.5, ~4 nm bandwidth) and a polarization-dependent isolator (ISO) for realizing the STML operation.”

(Supplementary Information 1)

Comment 5: *What is the differential group delay for the parabolic fiber used between different propagating modes and is this an issue after thousands of roundtrips?*

Authors' response: The second-order dispersion (β_2) of the first 10 propagating modes of the GRIN fiber (i.e., these used in our numerical studies, **Supplementary Information 9**) are provided in **Fig. R11**. As discussed in Ref. [*Science* 358, 94–97 (2017)] and [*Nat. Phys.* 16, 565–570 (2020)], the modal and chromatic dispersions can be counteracted by strong spatial and spectral filtering (i.e., the bandpass filter mentioned before) in the multimode fiber laser, leading to spatiotemporal mode-locking between the transverse and longitudinal modes.

To address the reviewer’s concern, we have added the second-order dispersion of the GRIN fiber to the supplementary information of the revised manuscript, i.e., **Fig. S15**.

Fig. R11. Second-order dispersion of the GRIN fiber. Here, the first 10 propagating modes are considered at 1063 nm.

Comment 6: *The authors should also mention relevant works in this area like: M. A. Eftekar et al, Nature Communications 10, 1638 (2019), Optics Express 25, 9078–9087 (2017), and Lopez-Galmiche et al, Optics Letters 41, 2553-2556 (2016).*

Authors’ response: We have cited these references mentioned by the reviewer and correspondingly discussed them in the revised manuscript:

“So far, various nonlinear dynamics in multimode fibers have been studied, such as accelerated nonlinear interaction⁷, octave supercontinuum generation^{8–10}, dispersive wave generation^{7–9}, spatial beam self-cleaning⁵, intermodal nonlinear mixing¹¹ and self-organized instability¹², to

name a few.” (Introduction, where Ref. 7 is [*Nat. Commun.* 10, 1638 (2019)], Ref. 9 is [*Opt. Express* 25, 9078–9087 (2017)], and Ref. 10 is [*Opt. Lett.* 41, 2553-2556 (2016)])

Again, we really appreciate the reviewer for his/her valuable comments on improving our manuscript.

We have carefully addressed all the concerns, and hope it merits the publication requirements.

Reviewer #3: *In this manuscript, the authors report a new experimental setup to characterize the pulse-to-pulse dynamics of spatiotemporal mode-locked lasers. By combining the dispersive Fourier transform and time division multiplexing, they observe a wide range of ultrafast dynamics. The results are new and interesting. I will support its publication if the authors address the following comments.*

Authors' response: We thank the reviewer for the positive comments on our manuscript. The responses to the comments are presented as follows.

Comment 1: *In Fig. 2b and c, I suggest that the authors display the vertical axis with frequency (or wavelength) units if possible. It would make it easier to read the spectral spacing of the interference fringes.*

Authors' response: We appreciate the reviewer for his/her careful studies on our results. All the results of the time-stretch (i.e., DFT) measurements in this work were plotted with double Y-axis, i.e., wavelength (right) and time (left) in **Fig. 2b,c**, which has also been adopted in the prior work [*Nat. Photon. 10, 321–326 (2016)*]. Plotting with both wavelength and time axes can facilitate the understanding of the wavelength-to-time mapping of the photonic time stretch. Please also note that, in **Figs. 2b,c** and **3b,c**, the birth of the STML solitons starts from the quasi-CW regime, in which the photonic time stretch is not valid [*Nat. Photon. 10, 321–326 (2016)*], as discussed in **Supplementary Information 4** of the revised manuscript. For this reason, we place “Time” on the left Y-axis that locates close to the quasi-CW regime of the STML birth, while “Wavelength” on the right Y-axis that locates close to the mode-locking regime. In **Figs. 4–6**, on the other hand, their evolutions are all in the mode-locking regime, and thus we state “Wavelength” on the left Y-axis to differentiate them from that of **Figs. 2b,c** and **3b,c**.

To address the reviewer's concern, we have elaborated it in the revised manuscript:

“b. Corresponding spectral evolutions. To facilitate the understanding of the wavelength-to-time mapping, the results of the time-stretch measurements in the birth stage are plotted with double Y-axis, i.e., time (left) and wavelength (right), which has also been adopted in the prior work²³.”

(Caption of Fig. 2)

“Please note that, different from the birth stage, here the time-stretch spectroscopy is valid for the whole evolution in the established stage, and thus the “Wavelength” axis is moved to the left to differentiate it from that of the birth stage, the same case for Figs. 5 and 6.” (Caption of Fig. 4)

Comment 2: *The inset in Fig. 2c should display units on the axes.*

Authors' response: We have modified this figure in the revised manuscript.

Comment 3: *“In the QSML state [...] the shockwave is gradually disappeared, [...]”. I believe that it should read “is gradually disappearing” or “gradually disappeared”.*

Authors' response: We have corrected it in the revised manuscript.

Comment 4: *“After the last QSML [...] enters the continuous MPML state.” I think that the term continuous should be removed as it can be mixed up with continuous wave operation.*

Authors' response: We have removed it in the revised manuscript.

Comment 5: *I found the sentence “In addition to the ability of real-time [...]” confusing as the experimental setup records the temporal dynamics without dispersing the signal (Fig. 1a). Moreover, a*

narrow bandwidth pulse could also have relatively large intensity, especially in Q-switching operating regime.

Authors' response: We appreciate the reviewer for his/her insightful comment on improving our manuscript. To address the concern from the reviewer, we have removed this sentence in the main text, as well as the corresponding description in the supplementary information of the revised manuscript.

Comment 6: *Interference geometry should be replaced by interference pattern.*

Authors' response: We have replaced it in the revised manuscript.

Comment 7: *It is not clear to me what the inset in Fig. 3b is showing. I cannot resolve the interference fringes. Is it because the fringe spacing larger than the pulse spectral bandwidth?*

Authors' response: We are grateful to the reviewer for the valuable comment. As we have described in the main text of the original manuscript, i.e., “*The solitons in P_2 , on the other hand, present no obvious interference fringe in the spectral domain (inset of Fig. 3b)*” (lines 211 and 212), the inset of **Fig. 3b** indeed aims to illustrate the spectral evolution of P_2 without visible interference fringes, which is different from that of P_1 as shown in **Fig. 3c**. This can be attributed to the fact that the large temporal separation of the pulse pair P_2 leads to non-resolvable interference fringes in the time-stretch measurement.

To address the reviewer's concern, we have modified the corresponding description in the revised manuscript:

“The solitons in P_2 without strong binding, on the other hand, present no obvious interference fringe in the spectral domain (inset of Fig. 3b), which can be attributed to the fact that the density

of the interference fringes is beyond the resolving ability of the real-time spectroscopy.” (Page 10, paragraph 1)

“**b. Corresponding spectral evolution. In this case, different pulse clusters coexist in the same roundtrip, including soliton-molecule with strong binding (P_1), soliton pair without strong binding (P_2) and ordinary soliton (P_3). Inset shows the close-up of P_2 .**” (Caption of Fig. 3)

Comment 8: *The authors say that there is only intensity modulation in the time domain on the pulses in SG3, but it looks to me that the intensity of pulses in SG1 are also oscillating at a high frequency which correspond to the oscillations observed in Fig. 5g. Moreover, I would also expect the energy to be the same in the temporal and the spectral domain.*

Authors’ response: We thank the reviewer for these valuable comments on improving our manuscript. In the original manuscript, by saying “*In the temporal domain, the intensity modulation is only recognized in SG₃ (left panels of Figs. 5e,f)*” (lines 260 and 261), we mean that obvious intensity modulation at a low frequency is observed in SG₃, as shown in Fig. R12. We agree with the reviewer that the intensities of the pulses in SG₁ oscillate at a high frequency, as shown in Fig. R13, similar to that of Fig. 5g. The corresponding intensity integration evolutions in the temporal domain exhibit similar features as that of optical spectra (Figs. 5g,h), as shown in Fig. R14.

Fig. R12. Temporal evolutions of the pulses in SG_1 and SG_3 . **a.** SG_1 . **b.** SG_3 .

Fig. R13. Intensity evolutions of the pulses in the temporal domain, i.e., Fig. R12. Here, the curves have been vertically offset for better visualization.

Fig. R14. Intensity integration evolutions of the pulses in the temporal domain, i.e., Fig. R12. Here, the curves have been vertically offset for better visualization.

To clarify it, we have added these results to the supplementary information of the revised manuscript, i.e., **Figs. S12 and S13**, and modified the corresponding descriptions to avoid confusion:

“In the temporal domain, i.e., left panels of Figs. 5e,f, the intensities of the pulses in SG₁ oscillate at a high frequency, while a low frequency for SG₃ (Fig. S12). Similar properties are also recognized for their temporal intensity integrations (Fig. S13). In the spectral domain, interestingly, the prominent spectral breathing is visualized in both SG₁ and SG₃ (right panels of Figs. 5e,f). The evolutions of their spectral energies (Figs. 5g,h), again, oscillate at different frequencies, similar to that of their temporal intensity integrations.” (Page 12, paragraph 2)

“e,f. Close-ups of the temporal (left) and spectral (right) evolutions of a,b and c,d, respectively, as indicated in a–d. g,h. Energy evolutions in the spectral domain (right panels of e and f).” (Caption of Fig. 5)

Comment 9: *In describing Fig.6. the authors say that they observe strong soliton fission. However, I am not really convinced by the experimental results that this is the case. Soliton fission occurs when a fundamental soliton is ejected and temporally separates from a higher-order soliton due to a strong perturbation.*

Authors' response: We thank the review for this insightful comment. **Fig. 6** presents the MUST observation in the multipulse STML regime, in which more pulses coexist in the laser cavity. We agree with the reviewer that, the soliton fission, in contrast to the mechanism of multipulse mode-locking here [*J. Opt. Soc. Am. B* 27, 1978–1982 (2010); *Opt. Express* 27, 11435–11446 (2019); *Appl. Phys. Express* 13, 022008 (2020)], has a different physics [*Opt. Lett.* 32, 391-393 (2007)].

To address the concern from the reviewer, we have modified the corresponding descriptions in the revised manuscript.

“A low saturation intensity of the IDT function can allow multipulse STML^{18,19}, which results in fruitful multipulse dynamics.” (Page 5, paragraph 2)

“The internal soliton dynamics have also been studied for scenarios involving more 3D solitons, as shown in Fig. 6a.” (Page 13, paragraph 1)

Comment 10: *Figs. 6c and d are not discussed in the text.*

Authors' response: We appreciate the reviewer for his/her careful studies on our results. We have modified the corresponding description in the revised manuscript.

“In the multipulse cluster, in addition to the ordinary solitons, e.g., P_1 as indicated in Fig. 6b, a spectrally and temporally explosive soliton (i.e., P_2) is unexpectedly observed, as shown in Figs.

6a,c (also their close-ups in Figs. 6b–d). The spectral-temporal explosion of P_2 tends to gradually decay (Fig. 6b). Analogous phenomena of collision-induced explosions with pulsation decay features were obtained in the 3D numerical simulations (Fig. S22). We also notice in Fig. 6e that P_3 manifests snake-walking spectral evolution with weak spectral width variation (Fig. 6f), while its temporal evolution presents weak intensity modulation (inset of Fig. 6a), which can be attributed to the vibration effect of the soliton-molecule (Fig. S23).” (Page 13, paragraph 1)

Comment 11: *While I find the dynamics reported in the manuscript very interesting, I think that this work could benefit from a discussion from the authors on the specificities and limitations of this new MUST setup. For example, could it be improved by adding a time-lens setup?*

Authors’ response: We are grateful to the reviewer for this valuable comment. To address the reviewer’s concern, we have provided a table to compare the performance of MUST with other technologies for 3D lightwave characterization, i.e., **Table R3** (**Table S3** in the revised manuscript). In addition, we have correspondingly discussed the limitation of the MUST system and the possibility of applying the time-lens technology to improve the temporal resolution of the MUST system in the **Conclusions and discussion** section of the revised manuscript:

Table R3 Typical technologies for 3D lightwave characterization

Technology	If single shot	If burst mode	Temporal resolution	No. of continuous frames
Delay-scanning off-axis digital holography	No	No	NA [*]	1 ^{**}
TERMITES	No	No	NA [*]	1 ^{**}
SEA TADPOLE	No	No	NA [*]	1 ^{**}
STRIPED FISH	Yes ^{***}	Yes	NA [*]	1
STS-CUP	Yes	Yes	500 fs	60
CUST	Yes	Yes	0.1–5 ps	60
T-CUP	Yes	Yes	0.58 ps	350
ISIS CCD	Yes	Yes	10 ns	16
FRAME	Yes	Yes	200 fs	4

LIF-DH	Yes	Yes	88 fs	7
SS-FTOP	Yes	Yes	276 fs	4
MUST	Yes	No	12.5 ps and 47.8 ns ****	>260,000 *****

*Not provided in the paper.

**Multiple measurements are required for reconstructing the 3D structure.

***Single-shot measurement only for low repetition rate lasers.

****Here, the temporal resolution in the temporal domain is defined by the sampling rate of the real-time oscilloscope (80 GS/s), while the temporal resolution in the spectral domain is defined by the repetition rate of the laser (20.9 MHz), i.e., 12.5 ps and 47.8 ns, respectively.

*****Here, the number of continuous frames is defined by the repetition rate of the laser (20.9 MHz) and memory depth of the real-time oscilloscope (1 Gpts/channel)

“Although sampling the mode profile in three speckle grains can unveil not only the speckly-diverse spectral-temporal dynamics but also their pairwise interrelations without loss of generality at a moderate system complexity, simultaneously sampling more speckle grains might provide a more comprehensive landscape. To illustrate this ability, we have provided an alternative MUST setup for measuring more speckle grains (Supplementary Information 11), and interesting MUST dynamics have also been observed. In addition, further efforts are yet to be made towards developing single-shot 3D characterization technologies with large numbers of continuous frames for measuring pulse-to-pulse 3D dynamics and fully understanding the nature of their complexities. For example, full-field characterization technologies^{25,26} that leverage the time-lens technology⁶² can potentially be employed to improve the temporal resolution of the MUST system (Table S3). ” (Conclusions and discussion)

We hope these revisions could satisfy the reviewer and meet the publication requirements. Thank you very much for your attention and consideration to our paper. We are looking forward to your decision.

REVIEWERS' COMMENTS

Reviewer #1 (Remarks to the Author):

The manuscript: "Real-time multispeckle spectral-temporal measurement unveils the complexity of spatiotemporal solitons" by Yuankai Guo, Xiaoxiao Wen, Wei Lin, Wenlong Wang, Xiaoming Wei, and Zhongmin Yang;

The authors responded to all comments and addressed well the Reviewers' concerns, while also adding additional measurements and information. I have found their replies convincing and satisfactory. Particularly beneficial is the substantial review of the introductory part, now properly reflecting the content of the manuscript while providing the pros and cons of the MUST system in comparison to other previously developed approaches. I do not have any further comments. I believe the manuscript can be now considered for publication in Nature Communication.

Reviewer #2 (Remarks to the Author):

The authors have adequately responded to my technical comments. I therefore recommend publication of this article in Nature Communications.

Reviewer #3 (Remarks to the Author):

The authors made a good job the revisions. I also had a look at the replies they provided to other reviewers' comments and I found them pertinent too. I would suggest to accept the revised version publication.

Manuscript #NCOMMS-20-16407A: “Real-time multispeckle spectral-temporal measurement unveils the complexity of spatiotemporal solitons”

We thank the editor for organizing the review and the reviewers for their valuable comments.

Authors’ point-by-point responses to reviewers’ comments

Reviewer #1: *The manuscript: “Real-time multispeckle spectral-temporal measurement unveils the complexity of spatiotemporal solitons” by Yuankai Guo, Xiaoxiao Wen, Wei Lin, Wenlong Wang, Xiaoming Wei, and Zhongmin Yang;*

The authors responded to all comments and addressed well the Reviewers’ concerns, while also adding additional measurements and information. I have found their replies convincing and satisfactory. Particularly beneficial is the substantial review of the introductory part, now properly reflecting the content of the manuscript while providing the pros and cons of the MUST system in comparison to other previously developed approaches. I do not have any further comments. I believe the manuscript can be now considered for publication in Nature Communication.

Authors’ response: We thank the reviewer for the insightful comments on improving our manuscript in the first round of review.

Reviewer #2: *The authors have adequately responded to my technical comments. I therefore recommend publication of this article in Nature Communications.*

Authors’ response: We are grateful to the reviewer for the valuable comments on improving our manuscript in the first round of review.

Reviewer #3: *The authors made a good job the revisions. I also had a look at the replies they provided to other reviewers' comments and I found them pertinent too. I would suggest to accept the revised version publication.*

Authors' response: We really appreciate the reviewer for his/her constructive comments on improving our manuscript in the first round of review.